

# Design, optimization and validation of genes commonly used in expression studies on DMH/AOM rat colon carcinogenesis model

David Bars-Cortina[1,2], Antoni Riera-Escamilla[3], Gemma Gou[4,5], Carme Piñol-Felis[2,6] and María-José Motilva[7]

[1] Food Technology Department, XaRTA-TPV, Agrotecnio Center, Escola Tècnica Superior d'Enginyeria Agrària, Universitat de Lleida, Lleida, Catalonia

[2] Department of Medicine, Universitat de Lleida, Lleida, Catalonia

[3] Andrology Department, Fundació Puigvert, Universitat Autònoma de Barcelona, Instituto de Investigaciones Biomédicas Sant Pau (IIB-Sant Pau), Barcelona, Catalonia, Spain

[4] Molecular Physiology of the Synapse Laboratory, Biomedical Research Institute Sant Pau (IIB Sant Pau), Barcelona, Spain

[5] Universitat Autónoma de Barcelona, Bellaterra, Spain

[6] Institut de Recerca Biomèdica de Lleida Fundació Dr. Pifarré-IRBLLeida, Lleida, Spain

[7] Instituto de Ciencias de la Vid y del Vino (ICVV) (CSIC Universidad de la Rioja-Gobierno de La Rioja), Logroño, Spain

Corresponding author
Carme
Piñol-Felis, pinyol@medicina.udl.cat

## ABSTRACT

Colorectal cancer (CRC), also known as colon cancer, is the third most common form of cancer worldwide in men and the second in women and is characterized by several genetic alterations, among them the expression of several genes. 1,2-dimethylhydrazine (DMH) and its metabolite azoxymethane (AOM) are procarcinogens commonly used to induce colon cancer in rats (DMH/AOM rat model). This rat model has been used to study changes in mRNA expression in genes involved in this pathological condition. However, a lack of proper detailed PCR primer design in the literature limits the reproducibility of the published data. The present study aims to design, optimize and validate the qPCR, in accordance with the MIQE (Minimum Information for Publication of Quantitative Real-Time PCR Experiments) guidelines, for seventeen genes commonly used in the DMH/AOM rat model of CRC (*Apc, Aurka, Bax, Bcl2, β-catenin, Ccnd1, Cdkn1a, Cox2, Gsk3beta, IL-33, iNOs, Nrf2, p53, RelA, Smad4, Tnf α* and *Vegfa*) and two reference genes (*Actb* or *β-actin* and *B2m*). The specificity of all primer pairs was empirically validated on agarose gel, and furthermore, the melting curve inspection was checked as was their efficiency (%) ranging from 90 to 110 with a correlation coefficient of $r^2 > 0.980$. Finally, a pilot study was performed to compare the robustness of two candidate reference genes.

## INTRODUCTION

Colorectal cancer (CRC) is the third most common form of cancer worldwide in men (surpassed by lung and prostate cancer) and the second in women (overtaken by breast cancer). The incidence of CRC varies significantly between populations, Australia and New Zealand being the countries with the highest rate of new diagnoses, while the countries of western Africa have the lowest incidence (*Fact Sheets by Cancer, 2018*). In the United States, CRC represents the fourth most prevalent cancer with 135430 new cases diagnosed in 2017 and representing 8.0% of all new cancer cases (*Colorectal Cancer-Cancer Stat Facts, 2018*). In Europe, CRC is the second most common cancer in both sexes (*Ferlay et al., 2013*). In Asia, especially in the industrialized regions, the incidence of CRC has increased over the last decade due to the adoption of the western lifestyle (*Koo et al., 2012*). Interestingly, a similar situation is taking place in Eastern Europe, Latin America and the Caribbean countries (*Arnold et al., 2017*). Unfortunately, the global incidence of CRC is expected to increase by about 60% and it is predicted that, in 2030, more than 2.2 million new cases will be diagnosed and 1.1 million people will die from this disease (*Arnold et al., 2017*).

It is well established that lifestyle and especially eating patterns play an important role in the risk of developing cancer in the digestive tract (*Slattery et al., 1998*; *Pan, Yu & Wang, 2018*; *Kurotani et al., 2010*; *Willett, 1994*). Hence, several studies have focused on diet as the major strategy to counteract and prevent colon cancer (*Grosso et al., 2017*). Different animal models have been used to evaluate the effect of food components on colon cancer prevention. Such models of colon carcinogenesis can be divided into two broad categories: transgenic and chemically-induced (*Perše & Cerar, 2011*; *Femia & Caderni, 2008*). A classic example of a transgenic animal CRC model concerns adenomatous polyposis coli (*Apc*) gene mutations. Nevertheless, these animal models have been generated to study familial adenomatous polyposis (FAP) and hereditary nonpolyposis colorectal cancer (HNPCC) syndromes which only account for approximately 5% of all cases (*Perše & Cerar, 2011*; *Femia & Caderni, 2008*).

Dimethylhydrazine (DMH) and its metabolite azoxymethane (AOM) are frequently used to generate chemically-induced models for CRC (*Perše & Cerar, 2011*; *Megaraj et al., 2014*). These models share many similarities with human sporadic colon cancer since DMH/AOM colon carcinogenesis occurs as a multistep process. The stepwise development of CRC from dysplastic crypts, adenomas to carcinomas provides the opportunity to investigate and identify molecular alterations in each stage of tumour development (*Perše & Cerar, 2011*). Interestingly, genes that have been found mutated in human sporadic colon cancer have also been found to mutate in DMH/AOM-induced colon carcinogenesis (*Perše & Cerar, 2011*). Nutrigenomics, the study of the effects of food components on gene expression, is a broad approach used in CRC animal models. In this regard, a wealth of data addresses this issue through different dietary regimes (more than 100 results appear when searching for ''rat colon cancer diet gene expression'' in Pubmed). However, the data generated is often confusing when it is carefully evaluated. For instance, in some published articles, the accession number of the reference sequence used is not indicated or the qPCR is not well designed or described (i.e., the primer sequence is missing or contains mistakes, primers

are not specific). On the basis of the aforementioned data, in this study we provide a well-designed and described qPCR protocol according to the MIQE (Minimum Information for Publication of Quantitative Real-Time PCR Experiments) guidelines (*Bustin et al., 2009*) for 17 genes routinely studied in DMH/AOM CRC rat model (*El-Shemi et al., 2016*; *Kensara et al., 2016*; *Islam et al., 2016*; *Qie & Diehl, 2016*; *Rivera-Rivera & Saavedra, 2016*; *Rubio, 2017*; *Al-Henhena et al., 2015*; *Tan et al., 2015*; *Gamallat et al., 2016*; *Walter et al., 2010*). Moreover, we also analyse two reference genes commonly used in this carcinogenesis model.

## MATERIALS & METHODS

### Animals

All animal care and experimental procedures were in accordance with the EU Directive 2010/63/EU guidelines for animal experiments and approved by the Animal Ethics Committee at the University of Lleida (CEEA 02/06-16). The project approved (CEEA 02/06-16) allowed the performance of a parallel study, described briefly on Fig. S1. However, from the same project, a group of remnants healthy adult male Wistar rats weighing between 200 to 250 g and maintained in the animal facilities at the University of Lleida were used for primer validation as a necessary previous step to perform a gene expression study. The animals were housed in polyvinyl cages at a controlled temperature (21 °C $\pm$ 1°C) and humidity (55% $\pm$ 10% RH), maintained under a constant 12 h light-dark cycle. All the animals were fed with water and a standard diet for rodents (Envigo Teklad Global Diet 2014, batch 3201, Settimo Milanese, Italy) *ad libitum*. Three randomly-selected animals were sacrificed by intracardiac puncture after isoflurane anaesthesia (ISOFlo, Veterinaria Esteve, Bologna, Italy). Distal colon tissue (the most relevant region in CRC studies with DMH/AOM induced models) (*Megaraj et al., 2014*) was extracted and immediately frozen in liquid nitrogen and then stored at −80 °C until it was analysed.

### RNA isolation & cDNA synthesis

Tissue Lyser LT (Quigen, Hilden, Germany) was used as a tissue homogenizer (four cycles of 50 Hz for 30 s. with a 1 min. pause within each cycle). Total RNA was extracted using the Trizol$^{TM}$ Plus PureLink$^{TM}$ Kit RNA Mini Kit (Invitrogen, USA) following the kit instructions. RNA quantity and purity (260/280 and 260/230 ratios) were assessed with a ND-1000 Nanodrop spectrophotometer (Thermo Fisher Scientific, Waltham, MA, USA). Furthermore, the integrity of the total RNA obtained was evaluated through 1% agarose gel (*Derveaux, Vandesompele & Hellemans, 2010*).

Reverse transcription was performed with the Maxima H Minus First Strand cDNA Synthesis kit with dsDNase (Ref. K1682; Thermo Fisher Scientific, Waltham, MA, USA) according to the manufacturer's instructions ($\leq 5$ µg of total RNA as template and using 100 pmol random hexamer primer). The resulting material was diluted with nuclease free water (BP561-1; Fisher Scientific, Waltham, MA, USA) for the qPCR reaction.

### Primer pairs design

Primer pairs for seventeen different CRC related genes (*Apc, Aurka, Bax, Bcl2, β-catenin, Ccnd1, Cdkn1a, Cox2, Gsk3beta, IL-33, iNOs, Nrf2, p53, RelA, Smad4, Tnf α and Vegfa*)

and two candidate reference genes (*Actb and B2m*) were designed and evaluated for their suitability through a number of bioinformatics tools summarized in Fig. 1A.

Briefly, we selected the genes to be studied (from a literature search) and obtained their accession number. Then, the nucleotide sequence was retrieved from the NCBI Nucleotide database which is feed from genome assembly of Rnor_6.0 (https://www.ncbi.nlm.nih.gov/nucleotide/). Afterwards, if no previously published primers were used, we checked for their splice variants (through Ensembl Release 87) (*Zerbino et al., 2018*) (https://www.ensembl.org/index.html). In the case of spliced genes and also in order to avoid the presence of SNPs (single-nucleotide polymorphisms), multiple sequence alignment was performed to find common regions within the splice variants (https://www.ebi.ac.uk/Tools/msa/clustalo/). *Zerbino et al. (2018)* The selected sequence was transferred to Primer3Plus software version 2.4.2 (*Untergasser et al., 2012*) to pick up primers (http://www.bioinformatics.nl/cgi-bin/primer3plus/primer3plus.cgi). The technical parameters used in the design of the primers were based on *Thornton & Basu (2011)*. Finally, the primer pairs obtained were submitted to further *in silico* analysis in order to avoid strong primer secondary structures (through OligoAnalyzer 3.1, Integrated DNA Technologies; https://eu.idtdna.com/calc/analyzer) (*Owczarzy et al., 2008*), robust amplicon secondary structures (with the UNAFold tool, Integrated DNA Technologies; https://eu.idtdna.com/UNAFold?) (*Owczarzy et al., 2008*) and unspecificity (with the In-Silico PCR tool of the UCSC Genome Browser Database (https://genome.ucsc.edu/cgi-bin/hgPcr) (*Rhead et al., 2010*) and the Nucleotide BLAST tool (https://blast.ncbi.nlm.nih.gov/Blast.cgi?PAGE_TYPE=BlastSearch) (*Johnson et al., 2008*)). The primer pairs selected after these bioinformatics tool tests were acquired from the Sigma-Aldrich custom oligo facilities (Haverhill, UK).

## PCR reaction & empirical validation

PCR reactions were performed in a total reaction volume of 25 µl comprising 2.5 µl of 10X Dream Taq Buffer, 0.5 µl of dNTP mix (R0191; Thermo Fisher Scientific, Waltham, MA, USA), 0.5 µl of gene-specific primer pair at 10 µM, 2 µl of cDNA template, 0.625 U Dream Taq DNA Polymerase (EP0701; Thermo Fisher Scientific, Waltham, MA, USA) and filled up to 25 µl with nuclease free water (BP561-1; Fisher Scientific, Waltham, MA, USA). The PCR conditions used were 3 min of polymerase activation at 95 °C followed by 35 cycles of denaturation at 95 °C for 30 s, an annealing step at 57 °C (or between 51 °C and 61 °C in the case of a gradient) for 30 s and extension at 72 °C for 30 s. Final extension (72 °C) was performed for 5 min followed by an infinite 4 °C step.

After the previous *in silico* steps described above, all the primer pairs were submitted to further analysis (Fig. 1B). Although the specificity of a pair of primers and absence of primer dimers is assessed in a more sensitive way using the melting curve in the qPCR reaction, it has been also considered opportune to check it through PCR.

Primer specificity was assessed through conventional PCR followed by agarose gel electrophoresis in order to check that unique band with the expected molecular weight according to the amplicon size was obtained. The annealing temperature was set at 57 °C by default but, in some cases, an annealing temperature gradient was needed (see above).

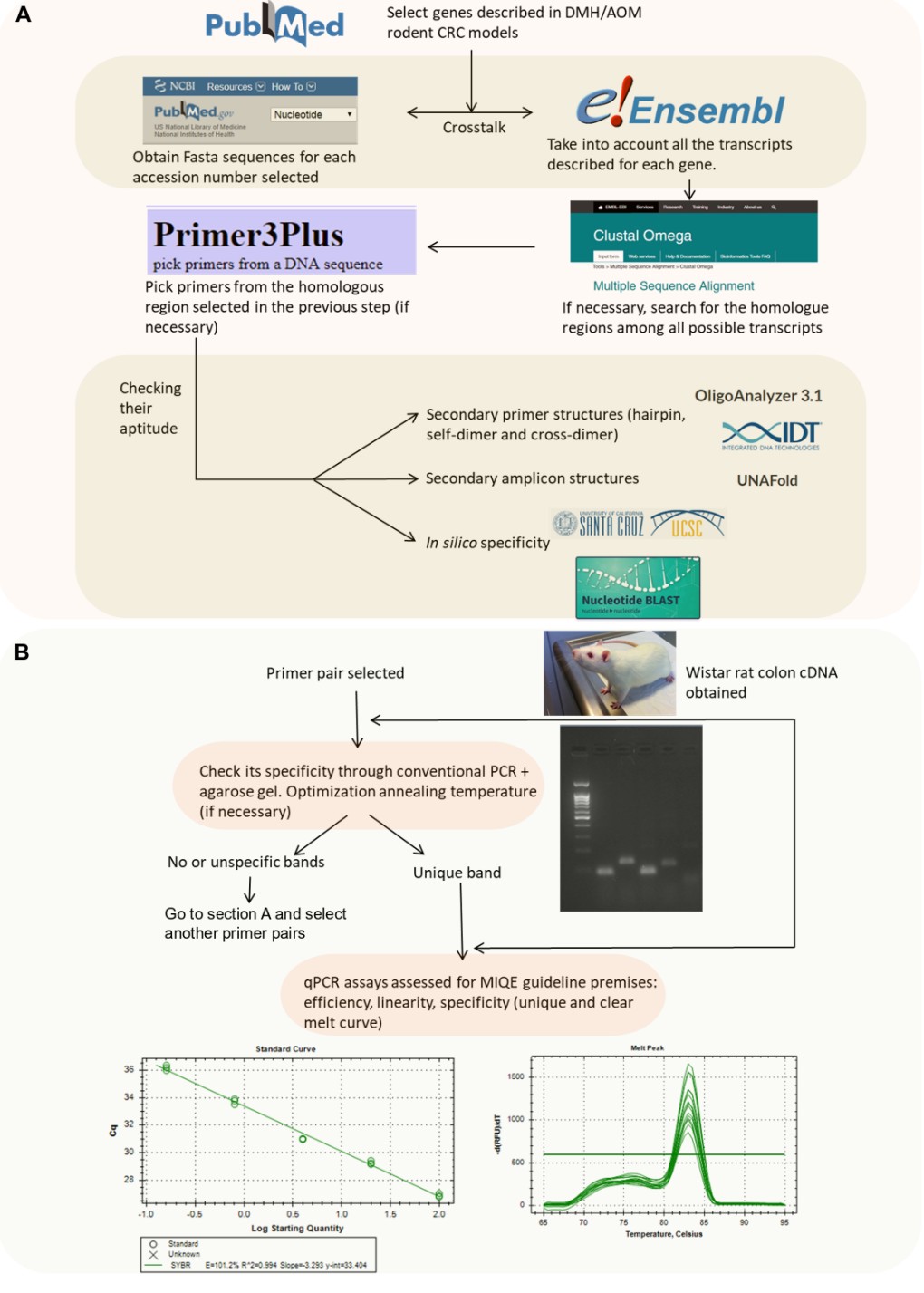

**Figure 1** **Flowchart indicating the strategy followed to design and validate the candidate primers.** (A) In silico validation flowchart. (B) Empirical validation flowchart.

## qPCR reaction, empirical validation and analysis

Real-time PCR reactions were performed in a total reaction volume of 20 µl comprising 10 µl of SYBR$^{TM}$ Select Master Mix (2X) (Thermo Fisher Scientific, Waltham, MA, USA), µl needed of each gene-specific primer (for every primer the concentration has been optimized from 100 nM to 400 nM), 2 µl of cDNA, and filled up to 20 µl with nuclease free water (BP561-1; Fisher Scientific, Waltham, MA, USA).

The qPCR reactions were carried out on a Bio-Rad CFX96 real-time PCR system (Bio-Rad Laboratories, Hercules, CA, USA) under the following conditions: 2 min of uracil-DNA glycosylase (UDG) activation at 50 °C, 2 min of polymerase activation at 95 °C, followed by 40 cycles of denaturation at 95 °C for 15 s and annealing/extension at the corresponding annealing temperature for 1 min. A melting curve analysis was done immediately after the qPCR analysis.

Once the unique band had been obtained in the previous PCR step, qPCR efficiency, linearity and specificity (unique and clear melt curve) were assessed taking into account (*Taylor et al., 2010*), and therefore the MIQE guidelines (*Bustin et al., 2009*). qPCR efficiency must be within a range of 90 to 110% and with a standard curve correlation coefficient ($R^2$) ≥0.98 (*Taylor et al., 2010*; *Kennedy & Oswald, 2011*). Each point on the standard curve was performed in triplicate. Whenever possible, the standard curve comprised three orders of magnitude. $C_q$ values >38 were not considered for data analysis due to their low efficiency (*Bustin et al., 2009*). Furthermore, in triplicate, no template control (NTC) was included for each primer pair in every run. The data resulting from the qPCR were analysed using the Bio-Rad CFX Maestro 1.1 software. Baseline correction and threshold setting were performed using the automatic calculation offered by the same software.

## Reference gene selection

The primer validation described in this paper is the necessary first step before to perform future relative gene expression studies using these primer pairs. In addition, in order to normalize the data, a reference gene choice is mandatory. The selection of an adequate reference gene is crucial because the expression levels of the reference genes may change between tissues and species and might be also influenced by experimental conditions of an experiment. Hence, for each experiment it is highly recommended to empirically choose the best reference gene for our study apart from a bibliographic search. As an example of this issue, and in parallel to the primers validation, we have conducted an experiment addressing the possible effect of dietary supplementation with a particular fruit (white- and red-fleshed apples) and cyanidin galactoside (the main anthocyanin in red-fleshed apples) on these genes in the early phases of rat colon cancer induced by AOM (Fig. S1). For this reason, two reference genes commonly used in DMH/AOM rat model experiments were selected and submitted to check their expression stability in the different experimental groups (Fig. S1). In detail, two distal colon from two rats per treatment group were analysed with three technical replicates each one. The amount of cDNA used in each reaction was 100 ng.

The stability (aptitude) of the candidate reference genes was evaluated with two software tools (web-based RefFinder platform: http://leonxie.esy.es/RefFinder/ accessed 08/05/2018) and Bio-Rad CFX Maestro 1.1. software, based on the geNorm algorithm).

# RESULTS

## Genetic material used

As stated in the previous section, three healthy adult male Wistar rats were selected randomly and sacrificed. The distal region of the colon was obtained and immediately frozen. The distal colon samples were pooled prior to total RNA extraction. The quality and quantity of the RNA was good (ratio 260/280 = 1.89, ratio 260/230 = 2.05, 186.6 ng/μl). Furthermore, the integrity of the total RNA obtained was evaluated through 1% agarose gel (*Derveaux, Vandesompele & Hellemans, 2010*). In all cases, 18S and 28S ribosomal RNA bands were clearly detected and no degraded RNA (illustrated as smear in the gel lane) was identified (pdf S1).

### *Primer design and validation through agarose gel*

The primer pairs detailed in Table 1 passed all the bioinformatics tests described in Fig. 1A. In particular, Table 1 specifies the nucleotide sequence of all primers from each gene studied (with their gene accession number); their map on mRNA rat genome (Rnor_6.0); their amplicon size; their annealing temperature used; and, if the primers were in-house designed or not.

Furthermore, as stated on Fig. 1B, a PCR + agarose gel has been performed in order to check that a single band with the expected molecular weight was obtained. Some examples of figures showing the agarose gel results are attached in pdf S2.

These primers were selected for further analysis through qPCR. On the contrary, the primers which do not pass some step in the validation process are shown in Table S1. The majority of the primer pairs (12, if we also consider the reference genes) were in-house designed. The remainder were from published intact sequences or, in some cases, were obtained after some *in silico* mismatch corrections (indicated as "Based on" in Table 1 and Table S1).

### *Primer validation through qPCR*

Through qPCR technique the primer pairs were submitted to an extra-control of their specificity because is a more sensitive analysis: melting curve. As can be verified in Fig. S2, a unique peak was detected for each primer pair, demonstrating their specificity and demonstrating the absence of primer dimers.

In addition, we also need to validate the qPCR assay, mainly qPCR efficiency. Table 2 summarizes the qPCR validation results obtained. In detail for each gene, the linearity range, lowest and highest Cq value used, qPCR efficiency (in %), coefficient of determination ($R^2$) and primer concentration used were detailed. Some examples of efficiency qPCR assays output are attached in pdf S2.

Furthermore, repeatability and reproducibility has been assessed (Fig. 2) from the expression analysis study mentioned previously (Fig. S1). In detail, the approach followed

**Table 1  List of the primer pairs validated.**

| Gene (Accession no.) | Primer sequences (5′–3′) | Gene region | Amplicon size | Ta | Reference |
|---|---|---|---|---|---|
| Actb* (NM_031144.3) | F: TCTGTGTGGATTGGTGGCT | Exon 6, CDS | 80 bp | 57 °C / 59.3 °C | In-house |
|  | R: TCATCGTACTCCTGCTTGCT | Exon 6, CDS |  |  |  |
| Apc (NM_012499.1) | F: ACTCCTTACTGCTTCTCACG | Exon 15, CDS | 114 bp | 57 °C | In-house |
|  | R: GTCCTTACTTTCTTTGCCCTTT | Exon 15, CDS |  |  |  |
| Aurka (NM_153296.2) | F: AGTGCTATCTGTCCATCAACC | Exon 8, 3′ UTR | 98 bp | 59.3 °C | In-house |
|  | R: ACCCGCATTTCCAGTCATCT | Exon 8, 3′ UTR |  |  |  |
| Bax (NM_017059.2) | F: AGAGGATGATTGCTGATGTGG | Exon 3, CDS | 93 bp | 57 °C | In-house |
|  | R: CCCAGTTGAAGTTGCCGT | Exon 4, CDS |  |  |  |
| Bcl2 (NM_016993.1) | F: GATTGTGGCCTTCTTTGAG | Exon 1, CDS | 232 bp | 59.3 °C | Based on *Zucchini et al. (2005)* |
|  | R: CAGGCTGAGCAGCGTCTTC | Exon 2, CDS |  |  |  |
| B2m* (NM_012512.2) | F: CCCACCCTCATGGCTACTTC | Exon 4, 3′ UTR | 157 bp | 57 °C / 59.3 °C | *Tan et al. (2015)* |
|  | R: GATGAAAACCGCACACAGGC | Exon 4, 3′ UTR |  |  |  |
| β-catenin (AF121265.1) | F:CAAGTGGGTGGCATAGAGG | Exon 8, CDS | 93 bp | 57 °C | In-house |
|  | R: ATGACGAAGAGCACAGATGG | Exon 8, CDS |  |  |  |
| Ccnd1 (NM_171992.4) | F: AGTTGCTGCAAATGGAACTG | Exon 2, CDS | 93 bp | 57 °C | Based on *Wu et al. (2012)* |
|  | R: TGGAGAGGAAGTGTTCGATG | Exon 3, CDS |  |  |  |
| Cdkn1a (NM_080782.3) | F: ATGTCCGATCCTGGTGATGT | Exon 1, CDS | 90 bp | 57 °C | In-house |
|  | R: GCTCAACTGCTCACTGTCCA | Exon 1, CDS |  |  |  |
| Cox2 (AF233596.1) | F: TGTATGCTACCATCTGGCTTCGG | Exon 7, CDS | 94 bp | 57 °C | *Peinnequin et al. (2004)* |
|  | R: GTTTGGAACAGTCGCTCGTCATC | Exon 7, CDS |  |  |  |
| Gsk3beta (NM_032080.1) | F: TGGGTCATTTGGTGTGGT | Exon 2, CDS | 95 bp | 57 °C | In-house |
|  | R: GGTTCTTAAATCGCTTGTCCT | Exon 2-3, CDS |  |  |  |
| IL-33 (NM_001014166.1) | F: TTCAGTCCTGCCCTTTCCTT | Exon 9, 3′ UTR | 84 bp | 57 °C | In-house |
|  | R: TGTGGTGCGTGCTCTTCT | Exon 9, 3′ UTR |  |  |  |
| iNOs (NM_012611.3) | F: CACCACCCTCCTTGTTCAAC | Exon 19, CDS | 132 bp | 57 °C | *Nergiz et al. (2012)* |
|  | R: CAATCCACAACTCGCTCCAA | Exon 19, CDS |  |  |  |
| Nrf2 (NM_031789.2) | F: GTGACTCGGAAATGGAAGAG | Exon 5, CDS | 83 bp | 57 °C | In-house |
|  | R: AGAAGAATGTGTTGGCTGTG | Exon 5, CDS |  |  |  |
| p53 (NM_030989.3) | F: GCAGAGTTGTTAGAAGGC | Exon 4, CDS | 138 bp | 57 °C | In-house |
|  | R: TTGAGAAGGGACGGAAGA | Exon 4, CDS |  |  |  |
| RelA (NM_199267.2) | F: TCACCAAAGACCCACCTCA | Exon 4, CDS | 81 bp | 57 °C | In-house |
|  | R: GTTCAGCCTCATAGAAGCCA | Exon 4, CDS |  |  |  |
| Smad4 (AB010954.1) | F: CCACCAACTTCCCCAACATT | Exon 5, CDS | 191 bp | 57 °C | *Kensara et al. (2016)* |
|  | R: TGCAGTCCTACTTCCAGTCCAG | Exon 7, CDS |  |  |  |
| Tnf α (NM_012675.3) | F: ACCACGCTCTTCTGTCTACTG | Exon 1, CDS | 169 bp | 59.3 °C | *Li et al. (2015)* |
|  | R: CTTGGTGGTTTGCTACGAC | Exon 3-4, CDS |  |  |  |
| Vegfa (ENSRNOG00000019598) | F: GACACACCCACCCACATAC | Exon 7, 3′ UTR | 141 bp | 57 °C | In-house |
|  | R: TCCAGTGAAGACACCAATAACA | Exon 7, 3′ UTR |  |  |  |

**Notes.**

*denotes reference gene.

Ta,  annealing temperature.
**Table 2  qPCR efficiency and correlation coefficient ($R^2$) obtained for each selected gene.**

| Gene (Accession no.) | Linearity range (ng cDNA) | Lowest Cq value | Highest Cq value | qPCR efficiency | $R^2$ | [primer], nM |
|---|---|---|---|---|---|---|
| Actb[*] (NM_031144.3) | 2 to 128 (Ta: 57 °C) | 17.4 | 23.1 | 108.9% | 0.998 | 100 |
| | 22.5 to 114 (Ta: 59.3 °C) | 16.4 | 19.2 | 90.5% | 0.999 | |
| Apc (NM_012499.1) | 2 to 128 | 23.7 | 29.4 | 108.5% | 0.998 | 100 |
| Aurka (NM_153296.2) | 9 to 243 | 28.6 | 33.2 | 108.8% | 0.998 | 200 |
| Bax (NM_017059.2) | 0.16 to 100 | 25.4 | 34.6 | 106.3% | 0.994 | 200 |
| Bcl2 (NM_016993.1) | 0.5 to 128 | 29 | 37.1 | 102.9% | 0.990 | 200 |
| B2m[*] (NM_012512.2) | 1.6 to 148.8 (Ta:57 °C) | 22.8 | 29.5 | 100.4% | 0.996 | 200 |
| | 2 to 128 (Ta:59.3 °C) | 23.1 | 26.8 | 108.7% | 0.997 | |
| β-catenin (AF121265) | 2 to 128 | 21.6 | 27.5 | 109.8% | 0.980 | 150 |
| Ccnd1 (NM_171992.4) | 22.5 to 114 | 22.1 | 24.3 | 108.5% | 0.997 | 100 |
| Cdkn1a (NM_080782.3) | 0.16 to 100 | 27 | 36.4 | 101.2% | 0.994 | 200 |
| Cox2 (AF233596.1) | 0.5 to 100 | 30.4 | 37.1 | 106.6% | 0.998 | 200 |
| Gsk3beta (NM_032080) | 1.56 to 100 | 23.8 | 29.7 | 106.8% | 0.997 | 200 |
| IL-33 (NM_001014166.1) | 2 to 128 | 25.5 | 31.1 | 110.0% | 0.996 | 100 |
| iNOs (NM_012611.3) | 20 to 338.8 | 30.8 | 35 | 100.0% | 0.990 | 200 |
| Nrf2 (NM_031789.2) | 0.16 to 100 | 23.1 | 32.3 | 106.6% | 0.996 | 200 |
| p53 (NM_030989.3) | 8 to 128 | 32.5 | 35.9 | 102.6% | 0.997 | 100 |
| RelA (NM_199267.2) | 0.16 to 100 | 24.9 | 34.1 | 103.0% | 0.997 | 200 |
| Smad4 (AB010954.1) | 6.4 to 100 | 23.7 | 27.6 | 104.5% | 0.993 | 400 |
| Tnf α (NM_012675.3) | 10 to 114 | 30 | 34.4 | 90.9% | 0.987 | 100 |
| Vegfa (ENSRNOG00000019598) | 0.16 to 100 | 22.5 | 30.1 | 106.9% | 0.992 | 200 |

**Notes.**
[*]denotes reference gene.
Ta, annealing temperature; nM, nanomolar concentration.

to explore these two parameters has been through the inter-run calibration (IRC) values obtained.

In our expression gene study, we analyzed each animal by duplicate for each gene of interest (GOI). The same sample of cDNA from one rat (aliquoted and stored at −80 °C)

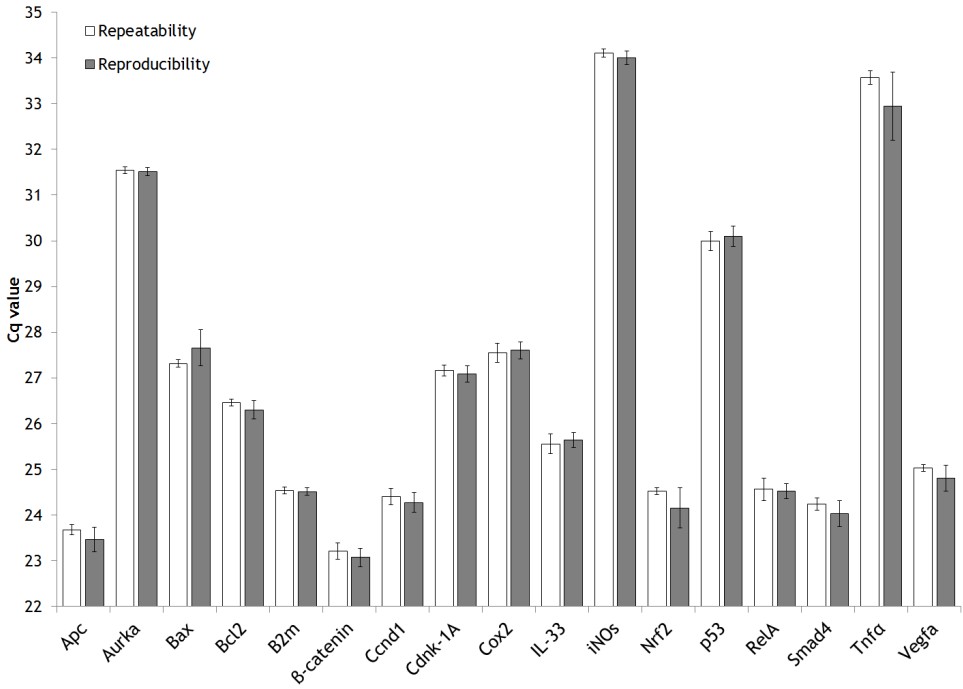

**Figure 2** **Intra-assay (repeatability) and inter-assay (reproducibility) precision of the validated genes.**
Bar graph showing the mean values of repeatability and reproducibility. Error bars representing the standard deviation (SD).

was used along the entire gene expression study of 17 GOI as IRC. The IRC it was also analysed by duplicate. The strategy of sample maximization method were used (*Hellemans et al., 2007*).

For each gene studied, the intra-run data of IRC has showed the GOI repeatability meanwhile the inter-run data of IRC has displayed their reproducibility. A figure clarifying this point is attached (Fig. S3).

In addition, the two reference genes demonstrated their ability to work properly at the two annealing temperatures (Ta) used (Table 2). This feature is desirable in order to normalize the results in qPCR studies because the gene of interest and reference gene should share a common Ta.

### Pilot reference gene validation

In this study, we selected two reference genes commonly used in DMH/AOM rat model experiments (*β-actin or Actb* and *B2m*).

The two reference genes were submitted to the same tests summarized in Fig. 1 as the other seventeen genes of interest (*in silico* design, specificity through PCR, qPCR assay validation and specificity through melting curve analysis). Nevertheless, for the reference genes, one further step was done: check their stability. As detailed previously in the 'Material and Methods' section, rat colon tissue samples from a parallel dietary intervention in the

AOM rat model experiment performed at the same time as this validation were used (Fig. S1).

In order to calculate the stability of the reference gene between the different experimental treatments (see details in Reference gene selection of 'Material and Methods'), we used two software tools: Bio-Rad CFX Maestro 1.1. software (based on geNorm algorithm (*Hellemans et al., 2007*)) and the output of the three software packages using the web-based RefFinder platform (*Xie et al., 2012*). In general, the two reference genes studied presented an analogous pattern with good expression stability (*M* values < 1.0) according to the geNorm algorithm, indicating excellent stability for both genes (pdf S3). Nevertheless, taking into account the RefFinder output (which considers the NormFinder and BestKeeper algorithms apart from the geNorm strategy) promotes the use of *B2m* towards to *Actb* gene (pdf S3). The gene with the lowest geomean value is viewed as the most stable reference gene. In detail, the software gave values of 1.19 and 1.41 for *B2m* and *Actb* genes, respectively. Accordingly, although the differences were minimal, *B2m* was established as the more appropriate reference gene in our long-term dietary study depicted on Fig. S1.

## DISCUSSION

### Primer design and validation

A correct selection of the primer pairs is a critical step for a qPCR experiment in order to obtain a specific amplification of the target gene. In addition, the primer design for SYBR® Green based detection needs to be more carefully done than for a classic TaqMan® assay since former interacts with double-stranded PCR products and may lead to ''false'' signal. Hence, the sensitivity of detection with SYBR® Green may be hindered by the lack of specificity of the primers, primer concentration and the formation of secondary structures in the PCR product. The formation of primer-dimers may register false positive fluorescence. However, this can easily be overcome by running a PCR melting curve analysis.

The primer pairs detailed in Table 1 passed all the bioinformatic tests (OligoAnalyzer 3.1 and UNAFold from Integrated DNA Technologies; in-silico PCR of the UCSC Genome Browser Database and the Nucleotide Blast tool) and a single band with the expected molecular weight was observed in the agarose gel.

In order to check whether the primers pairs designed were useful for qPCR analysis, we need to validate primer specificity again through the melting curve and also the qPCR assay, mainly qPCR efficiency, as stated in *Derveaux, Vandesompele & Hellemans (2010)* and *Taylor et al. (2010)*. One of the most common options to assess the specificity of the primer pairs is the melting curve. This determines whether the intercalating dye (SYBR green) has produced single and specific products. In this study we checked the melting curve for all the primer pairs and these all demonstrated their specificity as a unique peak was detected among the concentrations used in the standard curve in all cases. Therefore, no interfering and unspecific peaks were detected.

The determination of the efficiency of a qPCR should be among the first things to do when setting up a qPCR assay. The efficiency of a qPCR reaction is defined as the ability

of the polymerase reaction to convert reagents (dNTPs, oligos and template cDNA) into amplicon. Ideally, an efficient qPCR reaction achieves a twofold increase in amplicon per cycle (*Taylor et al., 2010*). In detail, PCR amplification efficiency must be established by means of standard curves and is determined from the slope of the log-linear portion of the calibration curve (*Bustin et al., 2009*). qPCR efficiency values must be within a range from 90 to 110% and with a standard curve correlation coefficient ($R^2$) $\geq$0.98 (*Taylor et al., 2010*; *Kennedy & Oswald, 2011*). As can be seen in Table 2, the efficiency of all the primer pairs designed ranged from 90.5% to 109.8% with an $R^2$ ranging from 0.980 to 0.999, which fulfils the requirements previously defined.

Although in the vast majority of the literature focused mainly on PCR efficiency (*Derveaux, Vandesompele & Hellemans, 2010*; *Taylor et al., 2010*; *Sun et al., 2015*), the establishment of the limit of detection (LOD) is also recommended by the MIQE guidelines (*Bustin et al., 2009*). Although we did not address this issue specifically, we indirectly came up against it. In some cases (e.g., *Aurka* and *p53* gene), apart from no signal detected at concentrations lower than 5 ng cDNA, the absence of a fluorescence signal and/or anomalous $C_q$ variation was detected within technical replicates in these low concentrations. Accordingly, the lowest standard curve concentration was increased in order to improve the qPCR efficiency.

## Pilot reference gene validation

Normalizing the data by choosing the appropriate reference genes is fundamental for obtaining reliable results in reverse transcription-qPCR (RT-qPCR). This process enables different mRNA concentrations across different samples to be compared (*Bustin et al., 2009*). Normalization involves the use of stably expressed endogenous reference genes in relation to the expression levels of the gene(s) of interest. However, the expression levels of the reference genes may change between tissues and species and might be influenced by pathological conditions and therapies (*Van Rijn et al., 2014*; *Dheda et al., 2004*; *Jacob et al., 2013*). Hence, an inappropriate choice of reference genes could lead to erroneous interpretations of results (*Dheda et al., 2005*). Therefore, the selection and validation of the reference genes is a crucial step before planning any expression analysis. In this study, we selected two reference genes commonly used in DMH/AOM rat model experiments (*β-actin or Actb* and *B2m*). To our knowledge, this is the first study to address the exploration of valid reference genes in rat colon tissue after dietary interventions.

In general, the two reference genes studied presented an analogous pattern with good expression stability (*M* values < 1.0) according to the geNorm algorithm, indicating excellent stability for both genes. Nevertheless, taking into account the RefFinder output (which considers the NormFinder and BestKeeper algorithms apart from the geNorm strategy) promotes the use of *B2m* towards to *Actb* gene. The gene with the lowest geomean value is viewed as the most stable reference gene. Accordingly, although the differences were minimal, *B2m* was established as the more appropriate reference gene in our long-term dietary study.

## CONCLUSIONS

qPCR is one of the methods of choice for gene expression analysis given its high sensitivity and because it works with very low nucleic acid concentrations. Nonetheless, there is a lack of qPCR validation information in the literature consulted. A lack of validation of the gene expression from the DMH/AOM rat model by qPCR is in line with the literature reviewed by *Jacob et al. (2013)*, who concluded that compliance with the MIQE guidelines continues to be an ongoing issue in the scientific community. Specifically, such essential information as the RNA integrity, the amount of cDNA, the linearity range and the efficiency of the qPCR is frequently missed.

In this study, with the aim of overcoming the lack of qPCR validation in the rodent CRC model, 17 rat genes related to human/rodent CRC were designed and validated following the MIQE guidelines (*Bustin et al., 2009*; *Taylor et al., 2010*). Furthermore, two reference genes commonly used in colon cancer studies were tested for their stability. Overall, this study provides a detailed list of 17 primer pairs for rat-related human/rodent CRC genes and demonstrates the proper stability of two reference genes in a particular dietary approach with the rat CRC model.

## ACKNOWLEDGEMENTS

We sincerely thank Isabel Sánchez and M Alba Sorolla (UdL SCT-Proteomics & Genomics Unit) for their advice on the empirical qPCR validation performed. The lead author would like to express gratitude to his family and Dra. Pilar Arbós Aixalà, Dr. Francesc Pujol Aymerich and Dr. Josep M. Pericay Hosta for helping him to continue their PhD journey positively.

### Funding

This study was supported by the Spanish Ministry of Education, Culture and Sport through the "Formación Profesorado Universitario (FPU)" grant awarded to David Bars-Cortina (FPU014/02902). The funders had no role in study design, data collection and analysis, decision to publish, or preparation of the manuscript.

### Grant Disclosures

The following grant information was disclosed by the authors:
Spanish Ministry of Education, Culture and Sport.
Formación Profesorado Universitario (FPU): FPU014/02902.

### Competing Interests

The authors declare there are no competing interests.

### Author Contributions

- David Bars-Cortina conceived and designed the experiments, performed the experiments, analyzed the data, prepared figures and/or tables, authored or reviewed drafts of the paper, approved the final draft.

- Antoni Riera-Escamilla authored or reviewed drafts of the paper, approved the final draft, technical revision.
- Gemma Gou prepared figures and/or tables, authored or reviewed drafts of the paper, approved the final draft, technical revision.
- Carme Piñol-Felis and María-José Motilva conceived and designed the experiments, contributed reagents/materials/analysis tools, authored or reviewed drafts of the paper, approved the final draft.

## Animal Ethics

The following information was supplied relating to ethical approvals (i.e., approving body and any reference numbers):

All animal care and experimental procedures were in accordance with the EU Directive 2010/63/EU guidelines for animal experiments and approved by the Animal Ethics Committee at the University of Lleida (CEEA 02/06-16).

## Data Availability

The raw data is available at the Open Science Framework: Bars-Cortina, David. 2019. "PeerJ Manuscript iD: 32888." OSF. January 4. http://www.osf.io/3wb9c.

## Supplemental Information

Supplemental information for this article can be found online at http://dx.doi.org/10.7717/peerj.6372#supplemental-information.

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
