# Peer review of "Design, optimization and validation of genes commonly used in expression studies on DMH/AOM rat colon carcinogenesis model"

_PeerJ, doi:10.7717/peerj.6372_

## Round 0.1 · original submission · Major Revisions

The article is well written, clear and could be useful for researchers working on the mentioned animal model. However, there are some criticisms that should be addressed by a revision. I am confident that the paper will be improved if you will answer to the comments and suggestions of the reviewers.

There is no need to show all the figures of the experiments as the second reviewer suggests but you should at least include the figure of the agarose gel loaded with the qPCR reactions to show the specificity of the primers and to indicate that the qPCR products have the expected molecular weights.

The second reviewer suggests to make a table with the website links, but it I think that it would be easier to include these informations in the methods sections.

The revised article should also include a better definition of the statistics and of the repeatability and reproducibility analyses.

·

Basic reporting

Table 1: It would be important to include for every standard curve the lowest and the highest Cq.
Why every gene has been tested for different amount of cDNA? Does it mean that under or above that amount of cDNA the curve is not anymore linear?

Table 2: It would be important to write for every primer where it maps on the cDNA (which exon, CDS or UTRs)

It would be nice to include and example of the agarose gel ran to check RNA integrity (rRNA 18S and 28S)

Rows 171-172: It is described the qPCR stating that the amount of primers used varies from 0.2 to 0.8 ul, depending on the concentration needed, that is stated on table 1, for each primer: since the relevant data is obviously the final concentration of the primer and not the volume, it is a useless and confusing information. The sentence should be removed or changed in “for every primer the concentration has been optimized, ranging from xxx uM to xxx uM, as indicated in Table 1”
Anyway, primers should have the same concentration.

Rows 286-289: it is not clear how the lowest standard curve was increased without increasing the minimum cDNA concentration

Figure S1 is far more detailed than what is needed. A simple table indicating the diet of the rats in the different groups would be enough. How it is, is just confusing

PCR reaction and PCR empirical validation should be in the same paragraph. Same for qPCR

the reference database for NCBI should be stated (Rnor_6.0), for future references

Row 133: “the nucleotide sequence was retrieved from the NCBI Nucleotide database”
(instead of PubMed Nucleotide database)

Experimental design

- The specificity of a pair of primers and absence of primer dimers is assessed in a more sensitive way using the melting curve in the qPCR reaction. The agarose gel is useful only to determine that the specific product has the attended molecular weight; this observation can be achieved running on the agarose gel the reaction after the qPCR. Thus, all the sections about PCR should be avoided.

- Row 209: This paragraph is a bit confused. It looks like the scope of the paper is to analyse the genes in rats with different dietary, while it is to assess the stability of reference gene USING rats on different dietary.

- Row 229: Why the samples were pooled for RNA extraction? To have more RNA? Moreover, it would be useful to know the total amount of RNA that has been extracted

- State in a clearer way how repeatability and reproducibility has been assessed (is the reproducibility of the triplicates in the same reaction? the same samples have been re-ran? Different retrotrascriptions from the same RNA? RNAs form different rats?)

- In FIG 2 some kind of test should be reported

Validity of the findings

- The first 4 primers sequences have been checked: primers of Actb, Apc and Aurka are designed on the same exon: using primers designed on the same exon enables to distinguish between cDNA and DNA, meaning that if DNA is a contaminant of RNA it will be amplified in the qPCR giving a product of the same length as cDNA. Ideally, primers should be designed either spanning to exons or on 2 different exons, in order to make sure that it is only cDNA that is being amplified. Otherwise, to avoid DNA contamination, one should consider to
1- treat the RNA with DNAse after trizol extraction
2- add another negative control in which RNA have been mixed with the RT mix NOT containing the enzyme (to assess that the RNA is not contaminated by DNA)

- Row 203: In the qPCR, together with a NTC negative control, negative controls for RT-PCR (no RNA) and for DNA (no enzyme) should be included

- The stability of the reference genes has been tested using rats that were following different dietary. The result is therefore very specific for this experiment and should be validated for any other treatment. This should be clearly stated

Additional comments

The publishing of primers for qPCR that can be used in rat model of colorectal cancer is surely useful and would save time to researchers working on the mentioned genes. However, it is very important to specify where the primers map on the mRNA. Additional negative controls should be included or suggested to be included. A part for a few paragraphs that I have indicated and that can be improved, the article is well written and clear. The conclusion that B2m is the best reference gene should be limited to the tested treatments.

I am not sure that it was necessary to sacrifice rats in order to test the efficiency of qPCR primers. It would have been sufficient to use a cell line. If the rats have been sacrificed for another project it should be stated.

Reviewer 2 ·

Basic reporting

- Please list a reference for the genes routinely studied in DMH/AOM CRC rat model and two reference genes commonly used in the carcinogenesis model (line 91-93).

- The paper lacks figure showing the results of the experiments performed.

- The authors should revise the “result” and "method" sections.
The method section should include a concise description of the used methods; the result section should include the description of the results, including also the figures of the experiments.

- The authors should include details regarding: how obtain the gene accession number, how to reach the Pubmed nucleotide database. Since the author is providing a standardized protocol, the paper would benefit of adding a table with the list of the useful website links and software cited.

- Please state how much RNA was used as starting material for the reverse transcriptase reaction and if the resulting material was diluted for the qPCR reaction.

-The authors explain that primers specificity was assessed by PCR followed by agarose gel electrophoresis in order to check that unique band with amplicon size was obtained (line164), please provide the data of this experiment. Did the author verify the product by sequencing?

- Please clarify the sentence “qPCR must be within a range of 90 to 110% ” (line 197).

- Please clarify the sentence line 199-200 “each point on the standard curve was performed in triplicate and covered all potential template concentrations that may be encountered in the future studies”.

- How the authors can quantify the cDNA used? Please explain (line 218).

- Figure S1 lacks a description, please modify accordingly.

- Line 227-229 state that the colon samples of 3 rats where pooled together prior the RNA extraction: the author should not refer to this experiment as triplicate.

- Line 260-263 the authors described that the unique pick of the primers tested, please show the data.

- Please clarify line 305-323: how the use of the software helped to define the stability of the reference genes? Which data were analyzed by using the listed software? Which data point/rat models were used? Please show clearly the data.

Experimental design

- The research question is well defined; methods are described with sufficient detail and information.

- The authors clearly describe the lack of reliable primers in previous literature, however it would be useful clarify lines 87-88. Based on what evidences the author states that “primer sequence is missing or contain mistakes, primers are not specific or the melting curve is too low”? (Lines 87-88)
Did the author tested the mentioned primers and verified the non-specificity? How the non-specificity was addressed? How the author defines a too low melting temperature?

Validity of the findings

- The authors describe nicely the process to design, optimize and validate reliable qPCR primers that will be used from future researcher in the field.
It is important include in the paper the figures describing the results of the performed experiment.

---

## Round 0.2 · accepted · Accept

The authors have satisfactorily responded to the reviewers questions and made corrections in the manuscript. The resulting article is robust, well written and clear, and will be useful for researchers working on DMH/AOM rat colon carcinogenesis model.

# ·

Basic reporting

no comment

Experimental design

no comment

Validity of the findings

no comment

Additional comments

The article is well written and clear in its scope and methods. It provides useful information for future studies in the field.
The authors answered to all my concerns and improved significantly the article.

Reviewer 2 ·

Basic reporting

The authors fully addressed all the points raised during the revision.

Experimental design

The authors fully addressed all the points raised during the revision.

Validity of the findings

The authors fully addressed all the points raised during the revision.

Additional comments

The authors properly addressed all the points raised during the revision.

I think the manuscript improved in terms of general clarity and description of the methodology and data.

Definitively this paper will be useful for scientist who will perform studies employing the listed validate primers.

Additionally, given the clarity of the experimental details and the access to figures's results, this paper will be helpful for scientist who will design new primers to study different genes non mentioned in the list.